# Adaptive Responses to Water, Energy, and Food Challenges and Implications on the Environment: An Exploratory Study of Harare

**Crecentia Pamidzai Gandidzanwa * and Muchaiteyi Togo**

College of Agriculture and Environmental Science (CAES), University of South Africa (UNISA), Florida 1710, Johannesburg, South Africa
* Correspondence: pamigandi@gmail.com

**Abstract:** Urban water, energy, and food (WEF) challenges are among the main barriers to poverty reduction and are some of the central targets of sustainable development goals (SDGs). SDGs seek to improve livelihoods in a sustainable manner through adequate and equitable distribution of the resources. In southern Africa, the scarcity of the resources has escalated due to increased pressure from urbanisation and climate change. This paper focuses on problems of the adaptive strategies that the communities are using in response to WEF challenges as well as the environmental implication of these choices. This article is based on qualitative research methods constituting interview guides administered to 6 city council officials, 2 NGO representatives, 35 households, and 1 Harare residents' association. Observations were undertaken, and review of secondary data was also done to collect information. Data were collated into a narrative, which was then exposed to qualitative content analysis. Findings reveal the use of underground water in both low- and high-income areas. Firewood and charcoal are preferred for cooking in the low-income suburbs, with gas dominating in the high-income areas. Mobile tuckshops, extensive backyard farming, and open-space agriculture were the prevalent sources of food. Inaccessibility and lack of affordability are some of the identified WEF challenges. Overdependence on underground water lowers the water table, increasing the ecological footprint. Uncontrolled urban agriculture exposes available water sources to pollution.

**Keywords:** WEF nexus; adaptive practises; sustainability; urban vulnerability; climate change; environmental problems

## 1. Introduction

Adaptive measures to respond to water, energy, and food (WEF) challenges are creating serious environmental problems in urban areas. This paper focuses on adaptive actions in response to WEF challenges as well as the environmental implications of these responses in Harare, Zimbabwe. Sustaining natural resources in the face of climate change and anthropogenic pressures is increasingly becoming a challenge in Africa [1–7]. Given that human factors are the main drivers of environmental changes globally, unplanned, unregulated urban growth and incorrect adaptation measures have the potential to create more pressures, generating new threats for the environment [1,2,5,8].

The 2007–2008 energy and food crisis led to the rise of the nexus concept [7]. This was debated and supported at the World Economic Forum (2011) and at the Bonn conferences (2011) and (2014) in recognition of the interconnectedness between WEF natural resources [9–12]. The Bonn Nexus Conference (2011) highlighted that the pressure on resources can result in shortages leading to WEF insecurity, which can have severe negative effects on the environment [5,9]. Currently, the WEF nexus has become central in the development and monitoring of SDGs and is also regarded as an SDG framework making implementation more efficient and robust [9,13]. Clean water, affordable energy, and zero hunger are some of the central targets of the SDGs [3,9,14,15]. WEF challenges and effects

of adaptive practices on the environment are undervalued, with developing countries lagging behind in acknowledging them [13,16–19]. Furthermore, adaptation to WEF and environmental implications in the context of vulnerability have been discussed mainly at a global and regional level and less at a local level [13,19,20]. If adaptive choices to WEF resources are not monitored and related vulnerability assessments ignored, there is potential to negate the green economy, a platform to fast-track SDG implementation [21–23]. This will also be a potential threat to global agreements, the Paris Agreement, Sustainable Development Goals, and the New Urban Agenda [20]. The New Urban Agenda, presented at the UN-HABITAT III Conference in 2016, was headed by the launching of the first urban SDG, SDG11, which seeks to "make cities inclusive, safe, resilient, and sustainable" [24].

The city of Harare has been struggling with the provision of WEF due to an increase in the urban population, a situation intensified by climate change. Wetlands have been under significant threat from anthropogenic activities, including property development and urban agriculture, threatening the city's water catchment processes [21,25]. Individual borehole drilling in the affluent suburbs of Harare has been accompanied by extensive digging of shallow wells in low-income areas, depleting the underground water supplies [26]. The country has experienced fuel and electricity shortages due to recurring droughts and fluctuations in rainfall patterns [21]. Zimbabwe is experiencing power cuts, load shedding, and brownouts [21,27,28]. The exclusion of climate change in urban policy, lack of comprehensive urban vulnerability assessments, and outdated master plans have hampered the achievement of sustainable development and resulted in failure to regulate development as evidenced by the increase in informal settlements [29,30] and the selling of bulky water [26]. In view of the challenges highlighted above and in light of the SDGs and the New Urban Agenda, sustainable and green adaptive measures and insights are required for informing policy to achieve smart cities.

This article places emphasis on household vulnerabilities and adaptive actions because this is where serious WEF challenges are faced, with the poor bearing the brunt of environmental risks [31]. It is at a household level that consumption patterns, a system of governance, performance of normal daily activities, and service provisioning affect the environment either positively or negatively [16,32–34]. Excessive exposure and limited adaptive capacity to climate change in cities is risky, in particular for the vulnerable poor in developing countries [18]. Studies revealed more vulnerability to heat-related distress mainly in densely built-up areas with low-income groups, exposing residents to, among others, energy and water shortages in Harare [35].

Literature has shown that people are aware of the challenge of climate change and the link with WEF shortages; however, they are less informed about environmental implication of their household activities and coping mechanisms [36,37]. It is not clear whether they are concerned about climate change and whether this impacts their decision to take mitigation measures [36]. It is also not clear whether urban residents are conscious that some of their daily activities are responsible for the emission of causing climate change [23]. In Kampala, Harare, and Nairobi, households rely on multiple sources of fuel, including charcoal and firewood [31,38,39]. Current research has established that greenhouse gas (GHG) emissions are also increasingly being driven by household energy services, including cooking [36]. The WHO guidelines call for a phase-out of fossil fuels and control in the use of wood fuel and other related sources of fuel to reduce GHG emissions for clean air and a stable climate [40]. Already, cities are responsible for 60% to 80% of global energy consumption and GHG emissions [41]. Controlling and reducing anthropogenic emissions can increase the amount of rainfall, positively contributing to water and food security [5,42].

Socio economic disparities and geographical location differences can be major drivers of climate change if not closely monitored [18]. In urban areas, the poor either lack or have difficulties in accessing potable water due to low incomes and long distances to the water sources [43,44]. Getting water is time consuming and expensive in Kampala, Uganda [31]. Household water insecurity at the household level is weakened by many trips to fetch water, unreliable water supplies, and long waiting times due to irregular supply and low

water pressure from water kiosks in Malawi [45]. In the southern part of Niger State, the majority of Nigerian respondents spent one hour to obtain water [46]. Most taps have run dry in most urban areas of southern Africa, resulting in the large urban population resorting to digging wells for domestic water [46]. Groundwater is often preferred for household domestic use because of its generally good microbial quality in its natural state, but it can easily be polluted if protective measures at the point of abstraction are not applied and well maintained [46]. Informal settlements are typically more prone to pollution due to reliance on contaminated sources and waste-management issues, which lower the quality of water sources, such as shallow wells [47,48]. Lucknow in India, Ribeiro Preto in Brazil, and Dar-es-Salaam in Tanzania are experiencing declining water tables due to uncontrolled private water-well drilling, and the in situ sanitation is causing high nitrate levels in shallow aquifers [49]. Most households in urban areas of developing countries are using shallow, unprotected wells, streams, dams, and ponds as alternative sources of water [26,39,44,45,50]. Peri-urban areas are providing most groundwater and surface water yet they are used as waste dump sites for the urban areas [51,52]. Water is fast depleting due to its invisibility, variability, increased abstraction, and often lack of regulation in urban areas, making management difficult [18,46,48,53–55]. Banning its abstraction to address depletion can trigger the informal market or increase the use of tanker water [44,55,56]. Water quantity problems in Harare have been attributed to the inadequate pumping capacity at Morton Jaffrey Works and Prince Edward Water Treatment Works [50,57], the two plants that supply the city.

Being connected to the grid does not necessarily lead to high consumption rate of energy, and access to electricity does not translate into availability [58]. Reliability of electricity supply, load shedding, power outages, and brownouts are major challenges in Africa [27,58–60]. Zimbabwe is one of the African countries where more than 50 percent of connected households receive power for not more than 50 percent of the time [27]. Some households that are connected to the grid never receive power. More than 30 percent of households in Uganda never have electricity despite being connected to the grid. Poor households in developing countries use traditional energy sources, including firewood, charcoal, kerosene, plant residue, and animal waste, because they lack access to or cannot afford basic modern energy services [61–63]. Low incomes and competing needs have pushed the majority of poor urban households into using solid fuels for cooking in southern Africa [63–65]. Wood and charcoal are used by most households without electricity in urban areas in southern Africa to meet their energy needs [46,63,64]. Charcoal is preferred for cooking because it is affordable, economical, convenient, and less bulky and releases less smoke [64,66]. Though it has a higher calorie value than wood, charcoal causes deforestation and pollution when burning indoors [66]. The use of traditional biomass for household domestic energy, affects the WEF nexus and intensifies the emission of black carbon, contributing to global warming [67]. Approximately 84% of urban households in Nigeria depend on diesel-powered/gasoline generators for electricity supply [68]. Diesel-powered generators have been associated with noise and air pollution, death cases at the point of fuelling (gasoline), and during overnight operations in Nigeria [60,69]. Lack of resources and expertise has hampered investments in off-grid systems [33,63,64]. The "fuel-stacking model" [70] suggests the use of multiple energy sources without altogether foregoing the old ones.

Urban food challenges result from income poverty because urban residents buy the majority of their food [4,39,71,72]. Among the barriers to accessing an adequate quality and quantity of food in India, like in many countries, are inadequate income; non-income dimensions, such as access to basic services, including water and energy; and access to social security, including social safety nets [73]. In Brazil, nutritional food highly depends on income; hence, the poor families buy low-nutrition food, cold cuts, and sausages, that is, ultra-processed foods they consume as their protein option [74]. In Nepal, the poor spend all their income on food but with limited or no access to nutritious food [72]. The poor urban population in Kenya basically eats to survive regardless of the quality of food [4]. In

Uganda, varying levels of food security in two small urban areas in Uganda experience the same challenges of food access and affordability [72]. Increasing economic challenges in Malawi (Blantyre), Zimbabwe (Harare), and Lesotho (Maseru) have resulted in increases in urban agriculture as a source of livelihood [72,75,76] in [41]. Among the factors that contribute to food security challenges in urban areas is climate change, overreliance on purchased food, inadequate sources of livelihoods, and lack of space for farming. In southern Africa, urban agriculture is not an effective means of reducing household food insecurity within the current operating urban regulations [41]. Residential areas have taken part of the agricultural land in Lesotho [75,77]. Urban agriculture can cause harm to the environment through increased pollution from fertilisers and pesticides that have washed away [26]. It can also act as a carbon sink, reducing GHGs [78]. Sugarcane burnings and drought have destroyed reforestation areas in the peri-urban areas of Brazil, and sugarcane growing has been associated with deforestation, habitat loss, soil erosion, GHG emissions, and impacts on water quality and availability [79].

WEF resources are interrelated, and impact on one sector affects the performance of the other sectors, hence the need to be tackled simultaneously [80–84]. In light of this, policymakers must focus their attention on understanding WEF scarcity challenges, adaptive responses, and environmental implications. This is significant, as it can make individual urban residents to be aware of their ecological footprints and to know the extent of the damage they will cause on the environment, which might influence them to take responsibility for their actions. Governments will have an opportunity to identify and integrate contextually suitable solutions in their environmental plans and policies to be more effective.

Action taken in response to climate change and its lead vulnerabilities of WEF shortages at a household level in an urban area depends on the households' adaptive capacity. Figure 1 is a simplified diagram of the pathways showing how urban vulnerability is expected to affect adaptive actions to WEF shortages and environmental implications. Climate change has a direct influence on water, energy, and food at the household level. It exerts added stress on urban areas through intense droughts, heat waves, and floods, compromising WEF supplies.

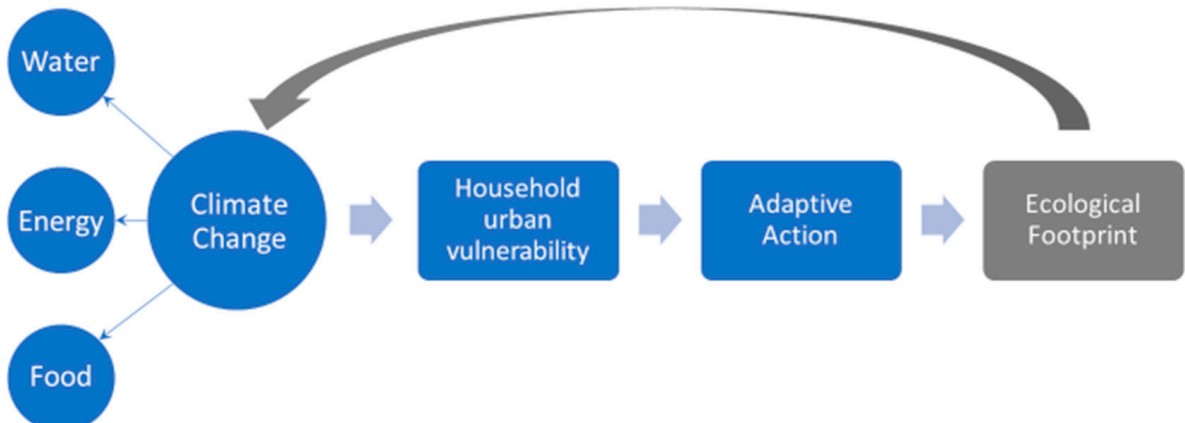

**Figure 1.** A conceptual framework showing the possible relationships between WEF challenges, adaptive actions, and the environment.

Existing scholarship on household adaptation to challenges of urban WEF often excludes consideration of the underlying ecological footprints of the community coping mechanisms, and the few references that mention the impacts of adaptation do so superficially [1,17]. Furthermore, no study in Zimbabwe has exclusively determined the extent of the effect of WEF adaptation practises on the environment in the face of climate change and urbanisation in Harare Metropolitan Province (HMP). An earlier study explored the WEF challenges experienced in Harare [85]. This paper investigates adaptive responses to

WEF challenges and their environmental implications. Rapid increase in urban population in southern Africa is stressing WEF resources with negative environmental impacts. In line with the Glasgow COP26 UN Climate Change Conference and the Paris Agreement, urban areas can be innovation hubs for sustainable transformative solutions to climate change. In this context, they can be sites to explore opportunities for household WEF nexus greening and for informing municipal regulatory policies where potentially environmentally harmful practices are concerned. This paper first explores the household adaptation practises to urban WEF challenges. It then discusses the environmental implications, drawing evidence from literature and research findings. In terms of structure, the paper consists of an introduction, methods, results, and discussion and conclusion sections.

## 2. Methods

### 2.1. Study Area and Data

Data were collected from five settlement types classified into informal (Hopley), peri-urban (Hatcliffe, Epworth), and formal high- (Budiriro, Mabvuku-Tafara), medium- (Zimre), and low- (Borrowdale) density settlement areas. Informal and peri-urban settlements represent areas with unplanned settlements with high population densities and extreme poverty rates, often prone to flooding, and often lacking water and energy connection and experiencing daily WEF challenges. Formal middle- and low-density settlements represent areas of high income, while formal high-density areas are of low income. The settlement types are distinguished based on the size of stands and socio-economic status, both of which have a bearing on adaptation strategies to WEF resource shortages. Borrowdale represents the leafy suburb of Harare with residents of high economic status. Mabvuku-Tafara was selected because residents have prepaid water meter taps, and there are also commercial water tanks that deliver water at designated places on specific days of the week at a given time gazette to the public. Budiriro was selected due to the recent outbreak of cholera attributed to the presence of unprotected shallow wells and the bursting of sewer pipes. Zimre was selected because of the prevalence of the green- and blue-tagged water tanks and the fact that the community has no municipal water. It buys water from private companies.

### 2.2. Study Design

This qualitative exploratory study aimed to explore household WEF challenges as well as local-level nexus adaptive approaches. The research was based on a case study of Harare. Case studies have the advantage of being intrinsic, allowing for in-depth examination of phenomena. Sampling of households in sites was purposive, with a focus on selecting those who use adaptation options, including fire/generators/solar or well/borehole/rainwater harvesting, etc. Where few people were using adaptation options, such as fire/refrigerators/solar or well/borehole/rainwater harvesting, etc., the researcher included everyone who used the options. Where there were many, systematic sampling was applied. However, the advantage of using a target was that it ensured that for each suburb, at least a few households were interviewed. Purposive sampling also ensured that a relevant and knowledgeable sample was selected and that the findings would be comparable to and possibly applicable in other contexts [86]. This type of sampling was done to identify if similar factors influence the prevalence of challenges in different residential areas as well as similarities in adaptive approaches. It was also done to enrich data and to improve sense-making through comparing and contrasting these areas. Qualitative data collection methods employed include interviews, document analysis, and observations. An extensive literature review on the subject was conducted. This included analysis of existing journal articles on WEF challenges and nexus approaches.

In reality, knowledge on environmental problems and impacts does not translate into sustainable environmental behaviour, nor does lack of environmental awareness imply a poor environmental practice [32] which [87,88] contradicts. No single factor affects a person's current behaviour or is sufficient to cause behaviour change [32]. It can be the urge to survive that more greatly determines households' daily decisions than environmental

concern. Environmental behavioural theories give an insight on why the communities' behaviour sometimes does not reflect environmental concern. The theories will assist in understanding the WEF-use patterns, choice behaviours, and ecological footprints.

Primary data were collected through in-depth interviews (IDIs) and focus group discussions (FGDs). The heads of departments in water, energy, and food in the city council, department of works, ZESA (Zimbabwe electricity supply authority), and WFP, UNICEF, and residence association were purposively selected and interviewed as shown in Table 1.

**Table 1.** Data selection techniques and designated respondents.

| Department | Designation | Data Collection Method | Total Interviewed |
|---|---|---|---|
| City council | Water distribution manager<br>Wastewater manager<br>Production manager<br>Principal nutritionist<br>Principal environmental officer<br>Department of works superintendent | IDI | **6** |
| ZESA | Tariff analyst<br>Engineer | | **2** |
| UNICEF | Climate change officer | | **3** |
| WFP | Programme Policy Officer—Urban | | |
| | Programme Manager—Urban | | |
| Residents Association | Harare Resident Association representative | | **1** |
| Community | Household heads/adult | | **20** |
| | Community | FGDs | **15** |
| **Total** | | | **47** |

**IDI**, in-depth interview.

In total, 47 respondents were interviewed: 6 city council officials—2 from ZESA, 3 from local NGOs, and 1 from the residents' association—and 35 community members. The idea was to interview at least one member from each department and identify more people through snowballing. City council officials and NGO officers working with the community in water and food distribution and renewable energy innovations participated in order to understand the communities' WEF coping mechanisms. A total of 35 Harare urban residents from diverse residential areas were involved in the research to capture varied problems of adaptive measures in response to WEF challenges in their communities. Quality and adequacy of the sample size were ensured through theoretical saturation. This is a point where collecting more data (e.g., by interviewing more people) does not generate any new information. Theoretical saturation was reached before all the purposively selected respondents could be interviewed, indicating that the sample size was more than adequate in generating data that satisfactorily addresses the aim of the research.

Predesigned, semi-structured interview guides with different themes tailored for different participants were used. From the key persons, interviews sought to obtain information on the strategies in place to help residents cope with WEF challenges, when they were implemented, how, their successes, and possible effect on the environment. During the in-depth interviews, the researcher had an opportunity to do a narrative survey of three adults above 70 years who had stayed in Harare for their entire life and were able to narrate the climate change trends and relate it to WEF challenges, adaptive measures, and environmental implications. These interview guides were refined, and probes were added to include emerging themes as the data collection proceeded. The interviews were audio recorded and transcribed. Extensive note-taking during the interview process was also done in case of problems with recordings. Two FGDs comprising of seven to eight participants of mixed gender were conducted in two settlement types, high- and low-

income areas, in an effort to identify challenges and problems of coping mechanisms. The FGDs were also meant to enhance data quality through methodological triangulation. Data from both interviews and FGDs were then collated to create a narrative.

The analysis of the data followed an inductive approach, which is commensurate with qualitative approaches [89]. In this case, generalisations are made based on observations. Induction helped to generate conclusions regarding household adaptive strategies. Because induction takes place at the empirical level, retroduction was then employed to go beyond the empirical level where observed events (adaptive strategies) are used to reach logical conclusions [89]. In particular, counterfactual thinking was employed to draw from existing experiences and social realities, in particular, knowledge about impacts of human activities (adaptive strategies) on the environment to abstract and think of what may not be but also what might be, which in this case, entailed the establishment of potential environmental impacts of existing adaptive measures.

Lastly, abductive reasoning was also employed to reconceptualise the data with the aim of revealing the connections and interlinkages between WEF challenges, adaptive actions, and the environment. Abductive reasoning helps to describe, interpret, and explain phenomena in a new contextual framework. It also helps to establish patterns and connections between elements, and therefore, it enables establishing connections and patterns that may not be observable in empirical data. The conceptual framework developed to theorise the relationship between climate change, household vulnerabilities to WEF challenges, adaptive behaviour, and environmental impacts (Figure 1) was used to recontextualise the findings to establish if the findings from the study area fit within such a frame.

## 3. Results and Discussion

### 3.1. WEF Shortages and Coping Mechanisms

Identified adaptive approaches to WEF challenges fall into two categories as shown in Table 2, that is, those that are conducive or have reduced impacts on the environment when compared to a "business as usual" scenario and those with negative environmental effects.

Table 2 shows coping mechanisms and possible implications on the environment. From the table, it is evident that households in Harare use multiple WEF strategies to cope with scarcities. Water adaptive options in response to water shortages include fetching water from shallow unprotected wells, boreholes, and local springs/dams/ponds. Alternative sources of energy that households were adopting due to shortages of electricity include use of gas, solar energy, firewood, charcoal, kerosene, and sawdust. The energy sources are highly variable, and the behaviour choice and usage patterns by households are unpredictable, as determined by fluctuating prices, accessibility, or personal preferences. Urban poor households are failing to shift from traditional sources of fuel to modern energy sources because they lack access to or cannot afford basic modern energy services. Coping practices in response to food shortages include greenhouse backyard farming, extensive open-space farming, and buying food from neighbouring countries. Harare residents are using a diversity of livelihood sources in response to food crises. This section discusses household experiences in the context of urban vulnerability in Harare. In particular, it explores household adaptive actions in response to WEF shortages and analyses the extent to which the practises affect the environment.

### 3.2. Household WEF Adaptive Practises, Behaviour Choice, Pattern, and Related Problems

Coping mechanisms to WEF challenges differed from settlement to settlement depending on context. Variations in socioeconomic factors and geographical location can be a source of environmental challenges [18]. These are significant factors in the discussion. Shortage of water from primary sources pushed the urban community to heavily rely on underground water. Table 2 shows that the low-income areas depend highly on unprotected, shallow, hand-dug wells and the high-income areas on individual boreholes. Most households in the low-income settlements had shallow, unprotected wells in their

premises or those of the neighbours. The wells are seasonal and usually run dry in the dry season. They usually lack the internal well casing during their construction, as this is expensive for the local residents. Most of the wells are protected by either covering with a metal lid or casted cement. In one informal settlement, ponds in cemeteries were used as water sources. Several dysfunctional manual hand-pumps were found in all study sites. Water from local springs/ponds/dams was used for general use by urban residents in low-income settlements. Individual electrified/solar boreholes backed by generators during power cuts were commonly found in high-income areas. In low-income areas, these are centralised and timetabled (e.g., opening at 6.00 a.m.–12.00 noon), with long queues, and water is given with a limited quota. In addition to these problems, power cuts also force residents to look for other alternatives. The green/blue/black water tanks placed on 4-m-high steel stands were a common feature mostly in middle- and low-density areas.

**Table 2.** Adaptive strategies to WEF shortages and their implications on the environment.

| Resources | WEF Adaptive Sources and Strategies (Inductive Reasoning) | Location Where Commonly Used (Inductive Reasoning) | Contribution to Environmental Change (Retroductive Reasoning | |
|---|---|---|---|---|
| | | | *Positive* | *Negative* |
| **Water** (*Bus-as-usual*: Individual/communal municipality tap water) | Individual borehole | FM, FL | Enhances and preserves underground water when planned and regulated | Depletes underground water when unplanned and unregulated |
| | Communal borehole | Info, PU, FH | | |
| | Unprotected shallow well | Info, PU, FH | | |
| | River/stream/dam/pond | Info, PU, FH | | |
| | Rainwater harvesting | Info, PU, FH, FM, FL | Sustainable | ———— |
| **Energy** (*Bus-as-usual*: Energy grid resources) | Charcoal | Info, PU, FH | | Contributes to increases in carbon emission |
| | Firewood | Info, PU, FH, FM, FL | | |
| | Sawdust | Info, PU, FH | | |
| | Generator | Info, PU, FH, FM, FL | | |
| | Kerosene (paraffin) | Info, PU | | |
| | Solar energy | Info, PU, FH, FM, FL | Zero carbon emission | ————- |
| | Liquefied Petroleum Gas (LPG) | Info, PU, FH, FM, FL | Reduces carbon emission | ———— |
| **Food** (*Bus-as-usual*: Buying from the shop/minimal urban agriculture/ backyard farming) | Climate smart backyard agriculture e.g greenhouses, farming in tyres, mulching | Info, PU, FH, FM, FL | Sustainable | Loss of aesthetic values |
| | Rampant open space farming in the city | Info, PU, FH, FM, FL | ———— | Deforestation, erosion, wetlands destruction, pollution, loss of aesthetic values |

Info, informal; PU, peri-urban; FH, formal high-density; FM, formal medium-density; FL, formal low-density; *Bus-as-usual*, business as usual.

Tanker water trucks deliver bulk water to consumers in all settlement types. In low-income areas, water is delivered at designated locations at specific times and in limited quantities free of charge. For example, in one peri-urban area, tanker water is supplied four times per week to supplement the piped water, which comes once a week. In high-income areas, water is delivered at a cost especially in areas of high elevation, where the water table is far below the surface. One example of a bulk water company that provides water services to informal settlements is the Clean City, a pan-African Mauritius registered entity under

the Econet Group, whose vision is to find solutions for services, including safe and clean water and energy for sustainable development. One of the city council officials revealed that bulk water trucks are not supposed to be a long-term adaptation option to supply water but a stop-gap measure whilst new technology is being developed. However, it was reported that there is need to review the skills available for local authorities and to train people to come up with new ideas in technology. The city, at engineering level, seems to be lacking, according to one engineer. New ideas are not appreciated, and change is resisted.

The use of multiple sources of water for multiple domestic uses in an urban area can be a reflection of household urban vulnerability and challenges. Indeed, in an effort to respond to seasonal vulnerability, households relied more on multiple water resources. This was also acknowledged by the Joint Monitoring Programme (JMP), which views it as normal and common to use various water sources when there is a crisis [45]. Similarly, in Malawi, Kenya, and Uganda, the prevalent use of unprotected, shallow wells and ponds/rivers/dams as water sources was strongly influenced by seasonality and income [31,38,45]. This shows increased vulnerability and reduced household adaptive capacity in low-income areas. These findings concur with research results from studies in Harare [26,39,50,90]. This is also in line with findings in sub-Saharan Africa [44,56]. In urban areas, the poor either lack or have difficulties in accessing potable water due to low incomes and long distances [43,44]. In similar studies, obtaining water in Kampala, Malawi, and the southern part of Niger State was reported to be time-consuming expensive, individuals take many trips to fetch water and the water supplies are unreliable [31,45,91].

Results of the study revealed that electricity was unreliable and inadequate or even non-existent in some areas. This forced urban residents to adopt multiple alternative sources of energy for multiple domestic uses. According to the results in Table 2, gas is used mostly in high-income areas compared to low-income settlements. Firewood is used for cooking in all settlements. The complex nature of energy use is clearly illustrated using the energy-utilisation patterns for low-income areas. In low-income areas, when the price of gas increases, most households shift gradually to firewood. Findings revealed that among those who are able to afford gas in low-income areas, some have leaking, unstandardised, refurbished tanks that produce smoke when cooking. These are not safe, and the gas does not last. Sawdust-burning stoves for cooking and warming the house coexist with gas stove and solar for lighting, most often in the same room. In some houses, it is common to see each room with an independent and different way of providing the necessary energy services. In the peri-urban and formal high-density areas, it was reported that generally those who can afford high tariff charges and are connected to the electricity grid can also afford to buy gas during power outages as well as when gas prices are reasonably low. On-grid households with several tenants sharing a single meter use firewood for cooking, reserving electricity for lighting and charging phones due to high tariffs. Rates beyond 300 units (the first 300 kWh of power are cheaper and bought at the beginning of each month. For fairness, all tenants should contribute towards buying for each month and then share the total units equally. From then on, each tenant should buy independently when they have exhausted their first share which is exponentially unaffordable. Irrespective of grid-connection, households in low-income settlements rely mostly on firewood in the form of small *tsotso* (twigs) stoves and sawdust stoves for cooking. Firewood is reported to be expensive and inaccessible; hence, the stoves are considered extremely economical for low-income households. Often, minimal firewood is used in combination with a special type of disposed plastic waste in order to save firewood. Two respondents reported:

> "We use barks of orchard trees or cut down trees combined with disposed '*zvigubhu*' (plastic containers) for cooking."

> "We used to fetch wood from neighbouring farms, but there are no longer farms, so we buy firewood from the market."

Charcoal is mainly reserved for boiling tough, heat-intensive food, such as beans and *maguru* (offals). Small charcoal stoves burn for longer hours even overnight indoors with

ventilation. Charcoal is bought cheaply from neighbouring countries, such as Mozambique and Zambia.

Solar energy is used mainly for lighting, television/radio, and charging phones in all settlements. Differences in the use of solar energy between the high-income and the low-income settlements depends on the nature of investment. In high-income areas, there is a standard set up of the solar system with expertise. In low-income areas, phones could be seen being charged directly on batteries, and solar lamps often haphazardly hang with loose connections. Recently, there has been a shift from using candles and kerosene lamps for lighting to a low-cost solar lighting package comprising of solar lamps only. Candles and kerosene are reported as more expensive than solar lamps since they are replenished more often. In all the settlements, most of the residents reported the desire to move towards renewable energy if funds permit.

Similar studies done in Africa revealed that some households connected to the grid could not utilise electricity due to lack of affordability and power outages [58,59]. For example, findings in Harare revealed that households experienced load shedding and power outages. Illegal connections were also common. Results are also in line with findings in Zimbabwe [39,64] that energy poverty was prevalent among the low-income groups. The findings concur with findings in Kenya [66] and the "fuel-stacking model" by [70]. In Kenya, charcoal is preferred mostly due to less pollution, high calorific value, and because it lasts longer, especially when used in improved cooking stoves [66]. Results are also similar to findings by other authors in developing countries, whereby transition in types of energy is not only a result of income and prices of fuel but also attributed to distance to fuel source, geographical location, personal preferences, and other factors [54,65,92]. The magnitude and extent of influence of these factors on a household's choice of fuel type vary across different fuel types [63,93]. Results also conform to the theory of [32] that behaviour choice can be due to contextual factors rather than environmental concern.

Grocery shops in Harare were no longer able to provide affordable foodstuffs to the urban population. The urban residents, particularly in the low-income areas, could not afford to buy groceries from shops. Respondents in the low-income settlements reported meagre salaries and income. Backyard farming, which used to supplement vegetables, was hindered by water shortages due recurring droughts. Mobile tuckshops were preferred for credit facilities and were within short distances. Groceries were also reported to be cheaper in these tuckshops compared to distant supermarkets. Urban residents also preferred crossing the border to neighbouring countries, such as South Africa or Mozambique, to buy groceries in bulk. However, one respondent revealed:

> "The restrictive laws on buying food items have stopped us from crossing the border, and food here is expensive."

Larger trucks were also used to deliver foodstuffs from neighbouring counties at a cost. There was a decline in quantities of food that comes from rural areas due to drought and poor harvests. Open-space farming also declined due to the expansion of the urban built-up area. Few people were working in neighbouring farms, as peri-urban farms were converted to residential areas.

Findings revealed that NGOs sometimes intervene in times of food crises by providing in-kind or cash transfers. In-kind transfers is a type of public spending to help specific populations in the form of specific goods and services, which recipients get for free or at a reduced rate. A cash transfer is simply a payment from the government to help improve the lives of its citizens.). One official from an NGO revealed:

> "The urban population is highly vulnerable because of minimal coping mechanisms as compared to rural areas. Our pilot study in Epworth is to establish access to food. The food is there, but it's a question of access. WFP's role is not to substitute the salary or household income but to intervene through providing support. Residents do what they want to do, and we augment support to reduce the coping mechanism they are employing. We are helping people with monthly cash transfers to complement their coping mechanisms, so people will use the

money to purchase food and other basic needs. This helps the families to get through this tough period. We are working with the department of social welfare and other stakeholders. We continue adapting to changes by the government; for example, when the government banned the use of foreign currency, we also adjusted. We use the government matrix as a criterion for choosing the beneficiaries including the vulnerable, those who are chronically ill, etc."

Results of the research are similar to the findings of [75,77] that residential areas have taken part of the agricultural land in cities limiting urban agriculture. Findings are also similar to the results of [63], which reported challenges of food access and inadequacies. Inadequate income and non-income dimensions have been reported as among the barriers to accessing adequate quality and quantity food in some vulnerable households in India, Brazil, Nepal, Kenya, and Uganda [4,72–74].

### 3.3. Environmental Implications

Findings revealed over abstraction of underground water by all Harare residents as a coping mechanism to the intermittent supply or lack of water supply. Haphazard, unregulated, and unprotected shallow wells are increasing. No hydrological or geological assessments have been made for borehole or well construction in the areas. There is an increase in abandoned wells in low-income areas. It was observed that the abandoned pits were now used to dispose waste. The interviewed respondents reported fragmentation of environmental legislation and lack of law enforcement by the Environmental Management Agency (EMA). An example that was given is lack of accountability due to lack of communication between the Environmental Management Act, the Urban Council Act, and the Water Act.

Borehole drilling has led to the drying up of other wells in vicinity showing, and there is an impact on the water table. In one peri-urban area, a cooperative pulled resources together and drilled a borehole, resulting in all the other nearby shallow wells drying up, demonstrating groundwater depletion and a fall in the groundwater table. Members were allowed to connect piped water to their homes where they would use generators. Most households in the informal settlements had pit latrines built, defying the required specification in terms of distance of a pit toilet from a water source (according to the WHO, latrines should be located downhill and safe distance (usually 30 m) from the nearest water sources depending on hydrological and hydrogeological conditions wherever possible, with a minimum distance of six metres from the house). One representative of the Combined Harare Residents Association (CHRA) reported that most boreholes in Hatcliffe are decommissioned due to contamination. The city council official commented:

"The groundwater is getting depleted. Boreholes are now drilled up to 60 m or more as compared to the 40 m in Harare. There has been direct impact on wetlands. Boreholes in low-density areas must have specifications. On the ground, the borehole septic tank ratios are too high. Boreholes should be 100 m away from septic tanks, but neighbours have boreholes less than 100 m apart. Because of durawalls, one will never know the neighbours' distance from his."

It was reported that individual boreholes in low- and middle-income areas are not licensed. Licenses are offered to monitor groundwater discharge:

"We last saw the city council authorities coming to demand licenses when the economy was still dollarized."

Catchment councils have the mandate to give licenses/permits for borehole drilling. The city authorities only identify/survey an area for borehole drilling on request from the development partners. One city council official reported:

"Borehole proliferation in the city is not sustainable. It only diverts the local authority from finding the right solution. They dry up by October. Boreholes are provided by development partners. As city council, we do not recommend bush pumps. In providing boreholes, development partners need to consider drilling,

protection, maintenance, and casing. The city council intervenes in siting and maintaining the borehole. The major source of pollutants and contamination for borehole/shallow wells is surface water, not underground water. The problem is normally experienced at the onset of the rainfall season except in isolated cases."

Springs are perennial and provide water for most in the informal and peri-urban areas, with some running dry in the dry season due to the sinking water tables.

Several high-income households, especially in high, elevated areas, reported owning two water sources: one for domestic use and the other for landscape maintenance due to low water tables. The two boreholes safeguard against water shortages posed by low water discharges resulting from declining water tables during the dry season. The borehole for landscape maintenance captures runoff during the wet season through underground pipes and tanks and is reserved for the long dry season in the face of drought. One city council official questioned and doubted the safety of household water tanks when they are connected to the city council water or filled with water sold by bulk water trucks. It was reported that the Zimbabwe National Water Authority (ZINWA) recommends borehole water from the periphery of the urban areas to safeguard underground water in the city. Ironically, the incapacitation of solid waste management systems in Harare has led to peri-urban areas degenerating into solid waste-dumping sites. The same peri-urban areas also provide groundwater for mobile water trucks and street vendors. Water tanks are also used to store water from the city council. When water is treated from plants, it has chlorine residuals, which deplete to zero with time. There is, therefore, a chance of water contamination in the tank. Furthermore, water tanks perpetuate inequalities. One city council official reported:

> "Those with tanks have better access to water than those without or who cannot afford. Those with tanks at times fill up the tanks while bypassing the city council water meter, short-changing the service provider, and also getting more water than the others. By the time the water reaches other residents, the pressure will have declined, and there will be more air bubbles, and sometimes, it will be time up for water cuts. However, water tanks are good as a short-term response."

If the water tanks are not up to standard and are in different colours and shapes, they result in loss of aesthetic value in an urban area. Findings of the study also revealed tragic incidents that occur when galvanized steel stands give in to the weight of loaded water tanks or collapsing roofs, leading to serious injuries or death.

One official reported that wastewater treatment plans are very expensive and are upstream of the raw water. Due to financial constraints, the integrated water supply system has collapsed, and the underlying principles of treatment are violated, leading to water pollution. Urban agriculture is done without observing principles of the environment, for example, without following the recommended distance from the river; hence, most chemicals are washed into the rivers. Industrial waste is disposed untreated into rivers. The penalties charged for polluting are not prohibitive; hence, industries find it cheaper and therefore prefer to pollute and pay later rather than to adhere to the guidelines and avoid the fines. The biggest challenge is that pollution is a vicious cycle. The local environmental action plan seeks to address fundamental issues, such as the wetlands, but this has not yet been put into action. By the time it is going to be implemented, the wetlands might have been destroyed already. One city council official reported:

> "Wetlands act as sponges for capturing and cleansing water. But because of the loss of wetlands, pollution is going into rivers directly without cleansing."

There is also pressure regarding land requirements, and violations are taking place, with increasing political interference. In high-density suburbs, some shallow wells are dug where there are sewage leakages, resulting in raw sewage sipping in.

According to [47], shallow wells, if well managed and if located in appropriate geology, produce good-quality water. However, they are vulnerable to pollution from a variety of sources. These findings are consistent with results of other researchers [46,48,52] that

groundwater, being an "open resource" [53], is fast depleting due to competing needs and increased abstraction, resulting in surface water supplies shrinking or becoming polluted. Groundwater is mostly preferred for household use, in particular for drinking. This is because of its good natural microbial quality, but it can easily be polluted if there are no protective measures and regulations [46]. Groundwater can be a health hazard to people if it contains fluoride and arsenic elements [46]. These findings also concur with results by [46,51], who noted that there is substantial evidence that drought has resulted in groundwater and aquifer water quality reduction. In agreement, research has also confirmed that peri-urban water sources have become major providers of groundwater and surface water for growing urban needs and also dumping sites for solid urban waste and polluted water, impacting underground water quality [46,80]. Numerous studies have confirmed that the overburdening and haphazard use of groundwater and its invisibility and variability have increased difficulties in management despite institutional regulations that might be in place [48,53–55].

Charcoal is the main cause of deforestation, mostly due to unsustainable harvesting and inefficient production techniques [66] Charcoal processing is also wasteful, as 10–20% of raw wood is converted to charcoal during its production process. In addition, about 10–15% of charcoal is wasted along the supply chain as charcoal dust. Charcoal causes pollution and is a health hazard when burnt indoors. The use of wood fuel causes deforestation [46], and yet, vegetation act as a carbon sink as it absorbs carbon, which causes global warming. Deforestation can cause soil erosion and water pollution. The advantage of using charcoal over firewood is that charcoal has higher energy content per kilogram of fuel burned, is less bulky and easier to store and transport, and burns with less smoke [64].

Traditional energy sources increase the emission of GHGs into the atmosphere, which is a health hazard [36,42,94]. The burning of low-quality fuel for cooking and heating results in substantial emissions of short-lived climate pollutants (SLCPs) or intensifies the emission of black carbon, contributing to global warming [67]. Domestic activities, such as cooking, are among the major contributors of GHGs. The WHO advocates for eradicating fossil fuels and for a control in the use of wood fuel for a clean and safe environment [40]. According to [46,95], GHG emissions can lead to drought, affecting agricultural production. Households are often not aware of the negative effects of using biomass and of the advantages of using modern fuels [64]. Similar studies carried out in Nigeria by [60,69] revealed that diesel powered generators used as backups during power cuts as causes of noise and air pollution as well as death at the point of fuelling.

According to [78], urban agriculture has a potential to cause harm to the environment. It can increase pollution to water bodies through washing away of fertilizers and can contaminate food through the use of wastewater for irrigation [78]. One official from the city council reported:

> "Urban agriculture should never be allowed except in horticultural farms, for example, by encouraging drip irrigation. The current scenario is that urban agriculture is done haphazardly. Fertilizers are applied and rivers are polluted. There has to be a campaign to stop this, but due to political interference, it's failing."

Agriculture can reduce GHGs through garden management, which can act as a carbon sink [96]. Zimbabwe is a drought-prone country with a larger population depending on agriculture for livelihoods. Therefore, there is need to enhance water security. This is in line with the findings of [80,81] on the WEF system in drought-prone areas, who concluded that water scarcity was the main reason for vulnerability. Water is at the epicentre of development. Investments in water-saving technologies can improve water resources and reduce household water consumption. The exploitation of solar in the country reduces electricity demand and energy expenditure while improving the living standards for communities [93,97].

Further analysis of the results through abductive reasoning revealed the inter-connections that exist between WEF challenges, adaptive responses, and the environment. The conceptual framework developed in theory for potential modelling of these relationships

was employed based on the research findings. Based on this re-contextualisation of the findings (Figure 2), WEF challenges affecting households in urban areas emanate from the effect of climate change. The challenges increase household vulnerability and reduce adaptive capacity at the household level in an urban setting. For example, the global greenhouse gas emissions and rapid population growth in cities have exposed households to high and often unmet food shortages, energy poverty, and water demand. Findings in Harare revealed erratic rainfalls impacting urban agriculture. Coupled with financial problems, the vulnerable population rely on poor-quality food and firewood for cooking. Regardless of class, urban residents experience water shortages. Often, the poor urban areas struggle with (in)access to domestic water, contaminated water sources, and poor and unreliable water supply; they rely more on traditional energy sources, such as firewood and charcoal, leading to respiratory diseases and cannot afford to buy nutritious food [13,31,39,45,64,79,98]. It was evident in the findings that Harare residents rely on polluted water sources, such as dams and ponds, for domestic use; firewood and charcoal for cooking and warming the house, especially in winter; and also depend on non-nutritional food.

In order to perform their normal socially shared daily activities, such as cooking, gardening, and bathing, households use various unsustainable adaptive actions in the context of their vulnerability [31]. This includes rampant exploitation of ground water for domestic water supplies through digging of shallow wells, unregulated cultivation in open spaces to supplement food supplies, and use of traditional fuels, such as firewood and charcoal. This has serious environmental consequences, such as deforestation, retreating of the water tables to deeper layers, and air and water pollution. The problem is that climate change is causing WEF scarcity, especially for vulnerable households, resulting in adaptation measures that are creating environmental challenges that further contribute to causes of climate change (Figure 2).

It is important to note that high adaptive capacity does not always reduce vulnerability [16]. The high-income population fear high electricity bills and can rely on firewood and charcoal for warmth due to the nature of their households, which are expensive to heat [98]. The rampant drilling of boreholes, if not regulated and monitored, can destroy the aquifers. The use of diesel/petrol-powered generators for various domestic uses can pollute the air and can be a health hazard, especially when used indoors.

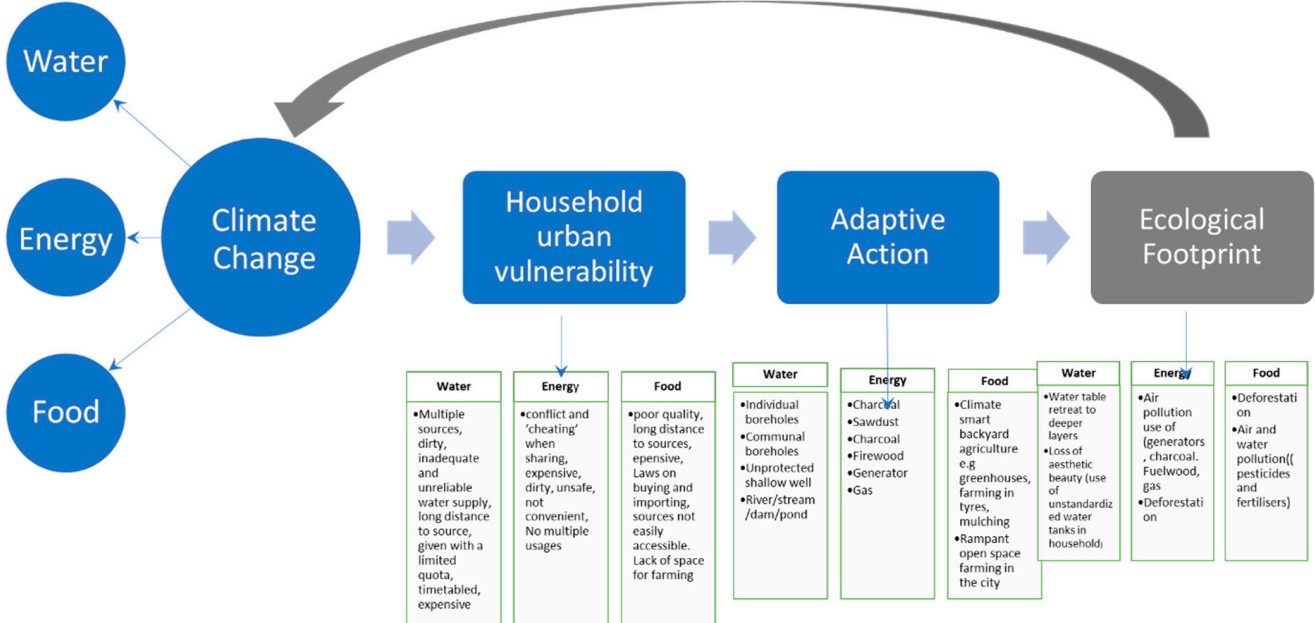

**Figure 2.** A structure showing the relationships between WEF challenges, adaptive actions, and the environment in Harare.

## 4. Conclusions

This paper discussed problems of WEF adaptive practises and environmental implications with the aim to promote green living in urban areas. Shallow, unprotected wells are common in low-income areas and individual boreholes in high-income areas. Overdependence on underground water can deplete the natural resource. Fuel types and use patterns vary across settlement types and are determined mainly by income, prices, access, and personal preferences and less so by environmental concerns. The use of firewood, kerosene, and charcoal as sources of energy can increase GHGs in the atmosphere. Food items are economically inaccessible due to competing urban needs. Open-space farming and greenhouse backyard farming are common. The country should aim to develop and improve its water system to boost its agricultural production. It also should move from traditional dirty fuels that pollute the environment to smart energy sources that are environmentally friendly.

**Author Contributions:** Conceptualization, M.T. and C.P.G.; methodology, M.T.; software, C.P.G.; validation, C.P.G. and M.T.; formal analysis, C.P.G. and M.T.; investigation, C.P.G.; resources, C.P.G.; data curation, C.P.G. writing—original draft preparation, C.P.G.; writing—review and editing, C.P.G. and M.T.; visualization, C.P.G. and M.T.; supervision, M.T.; project administration, C.P.G.; funding acquisition, C.P.G. with M.T. support. All authors have read and agreed to the published version of the manuscript.

**Funding:** The research was funded by the University of South Africa (PhD student 53338790 M & D Bursary).

**Institutional Review Board Statement:** The study was conducted in accordance with the Declaration of Helsinki, and approved by the Institutional Review Board (or Ethics Committee) of UNISACAES Health Research Ethics Committee (protocol code 2019/CAES_HREC/120) for studies involving humans.

**Informed Consent Statement:** Informed consent was obtained from all subjects involved in the study.

**Conflicts of Interest:** The authors declare no conflict of interest.

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
