# Peer review of "Adaptive Responses to Water, Energy, and Food Challenges and Implications on the Environment: An Exploratory Study of Harare"

_sustainability, doi:10.3390/su141610260_

Round 1

Reviewer 1 Report

The approach of the manuscript is interesting because it seeks to fill the lack of knowledge about the Urban water, energy and food (WEF) challenges In Southern Africa. However, the number of interviews carried out is very low. There is doubt about how strong the conclusion based on a limited sample is. The authors should reveal more accurately the representativeness of their sample, even though it is an exploratory qualitative study.

Author Response

Comments were addressed as follows:

The methodology section was elaborated.  After reaching  data saturation  47 respondents were interviewed in this qualitative research

The results and discussion section was worked on in order to tightly tie it with   empirical literature.

References were improved.

Reviewer 2 Report

The presented manuscript is focused on problems of great importance for the regions in concern. However, in order to be considered for publication, it requires major conceptual and methodological, as well as minor technical improvements.

My notes about the overall text are as follows:

  - please, check and remove the extra spaces;

  - please, rectify the incorrect transfer of words (for example rows 66, 200, 213, etc.);

  - in some places the references are in [ ], in others - in ( ).

I have the following questions and suggestions about the submitted manuscript:

  • The sentence within rows 18-19 “Data was collated into a narrative and content analysis was used for data analysis.” should be corrected.
  • Row 63-64: “Removing anthropogenic emissions” - they can be controlled and reduced, but they can`t be easily removed.
  • Please, check the term “copying mechanism” (rows 58, 170, 182, etc.), maybe you mean “coping mechanism”?
  • In Chapter 2. Methods, subchapter 2.1. Study area and data:

A very important issue of any scientific research is providing information/detailed description of the data sources, because normally such studies significant for society must be based on data quality assessment. Such an assessment procedure is typically a responsibility of the organization/database (if it is an open source) providing the sets of data used for the present analysis/research. Such information is sometimes available at the administrative authorities responsible for collecting, validating and reporting the data, e.g. on rural/regional/national/international level. Please, consider providing such information!

A clear description of the methods used for data validation/analysis/extraction (regarding the work done in the present research) is also needed.

  • In Subchapter 2.2: When sociological studies (row 139 - “interviews”, row 159 - “in-depth interviews”) are carried out for data collection, it would be good to present the so-called “questionnaire prepared/used” for the different sectors that are covered by the present study.
  • It is highly recommended to provide references for the data, presented in Table 2.
  • The data discussed in subchapters 3.2 and 3.3 should be thoroughly referenced and possibly analyzed (as minimum “benchmarking analysis”). The results can be summarized in well-structured materials (tables and/or graphical interpretations). Such interpretation would provide a clear overview of the major results and conclusions, drawn from the present research.
  • In Conclusions - the most significant observations deduced in the present study could be systemized herein. Why do you make this conclusion (row 471): “The country is characterised by an export dominated economy with a trade deficit…”, when your research deals with interviewed people in Harare about WEF (water, energy and food) and results from their answers?
  • Please, fill or remove (where it is not applicable) paragraphs within rows 475-504.

Author Response

Major conceptual  and methodological  as well as minor technical improvements were done. A conceptual framework later recontexualising findings was introduced in this research. Further analysis of results through abductive, inductive and retroductive reasoning was also  done. 

Round 2

Reviewer 1 Report

The improvements made are satisfactory.

Author Response

The reviewer was satisfied with all the addressed comments in Round 1 

Reviewer 2 Report

Dear authors, I have the following notes and recommendations about the article:

1) You use many times in the text the term “copying mechanisms” (for example page 3, row 3; page 5, row 2 after Figure 1; page 7, row 6 after Table 1; etc.). But there is a difference between “copy” and “cope with” (you wrote in this way/manner on page 7, the last paragraph, row 3). Please, check carefully and correct this term (copying mechanisms) in your text or specify this used term.

2) Please, unify the citation in the text, because you use different expressions, for example:

  - page 1, the last row - [76, 33, and 42];

  - page 2, row 3-4 - [9,17,95];

   - page 2, second paragraph - (58).

3) Please, check and modify the sentences on page 16, rows 9-12.

4) Table 2 - please, check and correct the added text in the headings.

Author Response

This manuscript is a resubmission of an earlier submission. The following is a list of the peer review reports and author responses from that submission.